# Field testing an "acoustic lighthouse": Combined acoustic and visual cues provide a multimodal solution that reduces avian collision risk with tall human-made structures

**Timothy J. Boycott**[1]*, **Sally M. Mullis**[1], **Brandon E. Jackson**[2], **John P. Swaddle**[1,3]

**1** Biology Department, William & Mary, Williamsburg, Virginia, United States of America, **2** Department of Biological and Environmental Sciences, Longwood University, Farmville, Virginia, United States of America, **3** Institute for Integrative Conservation, William & Mary, Williamsburg, Virginia, United States of America

* timothyjboycott@gmail.com

**Data Availability Statement:** All relevant data are within the paper and its Supporting Information files.

## Abstract

Billions of birds fatally collide with human-made structures each year. These mortalities have consequences for population viability and conservation of endangered species. This source of human-wildlife conflict also places constraints on various industries. Furthermore, with continued increases in urbanization, the incidence of collisions continues to increase. Efforts to reduce collisions have largely focused on making structures more visible to birds through visual stimuli but have shown limited success. We investigated the efficacy of a multimodal combination of acoustic signals with visual cues to reduce avian collisions with tall structures in open airspace. Previous work has demonstrated that a combination of acoustic and visual cues can decrease collision risk of birds in captive flight trials. Extending to field tests, we predicted that novel acoustic signals would combine with the visual cues of tall communication towers to reduce collision risk for birds. We broadcast two audible frequency ranges (4 to 6 and 6 to 8 kHz) in front of tall communication towers at locations in the Atlantic migratory flyway of Virginia during annual migration and observed birds' flight trajectories around the towers. We recorded an overall 12–16% lower rate of general bird activity surrounding towers during sound treatment conditions, compared with control (no broadcast sound) conditions. Furthermore, in 145 tracked "at-risk" flights, birds reduced flight velocity and deflected flight trajectories to a greater extent when exposed to the acoustic stimuli near the towers. In particular, the 4 to 6 kHz stimulus produced the greater effect sizes, with birds altering flight direction earlier in their trajectories and at larger distances from the towers, perhaps indicating that frequency range is more clearly audible to flying birds. This "acoustic lighthouse" concept reduces the risk of collision for birds in the field and could be applied to reduce collision risk associated with many human-made structures, such as wind turbines and tall buildings.

**Funding:** This work was supported by the Center for Innovative Technology's Commonwealth Research Commercialization Fund award MF18-029-En (https://www.cit.org/) to JPS, by the Animal Welfare Institute (https://awionline.org/), the Virginia Society of Ornithology (https://www.virginiabirds.org/), the Williamsburg Bird Club (http://williamsburgbirdclub.org/), and the Department of Arts and Sciences at William & Mary to TJB. The funders had no role in study design, data collection and analysis, decision to publish, or preparation of the manuscript.

## Introduction

Billions of wild birds die annually from collisions with human-made structures such as communication towers, wind turbines, power lines, and buildings [1–4]. These collisions are one of the largest sources of human-caused avian mortality world-wide. Such collisions can be a significant threat to species of conservation concern on a local scale [e.g. 5–7] and likely have larger scale impacts too, especially for migratory species [8]. Furthermore, losses in avian abundance can have functional costs for populations, communities, and ecosystems, for example by changing predator-prey dynamics [9–11], and local population declines can contribute to restricted geographical ranges and eventually extinction [12].

In addition to ecological effects, avian collisions are costly to numerous sectors of human society, including agriculture, travel, and renewable energy [13, 14]. For example, the threat and occurrence of collisions between birds and wind turbines, and the actions taken to mitigate this conflict, costs the United States wind energy sector hundreds of millions of dollars each year [15]. The widespread occurrence of collisions, affecting many avian taxa and types of human-made structures, renders them a prominent source of human-wildlife conflict. Furthermore, with continued urbanization, the incidence of bird collisions is projected to increase [16].

Unsurprisingly, there have been substantial efforts to reduce the incidence of bird collisions with human-made structures. Many of these efforts have focused on making structures more visible to birds [e.g. 17, 18]. Some methods have been successful, such as applying various types of markings on glass windows [19], altering the coloration of wind turbine blades [20], marking ground wires on transmission power lines [21], and adding dynamic lighting atop towers at night and in overcast conditions [22, 23]. But many efforts lack transferability and replicability among sites and fail in some situations [2, 4]. The goal of making structures more visible has often been informed by a human, not avian, perspective. However, birds view their worlds quite differently from humans [24, 25]; hence there is increasing realization that solutions to collisions need to incorporate knowledge about both the environmental context of hazards and the sensory systems of at-risk birds [24, 26, 27]. Here, we apply sensory ecology to generate a multimodal solution to reduce birds' collisions with tall open-air structures, such as wind turbines and communication towers.

The perceptual limitations of flying birds may be a factor that explains the prevalence of collisions with tall open-air structures [24, 25]. Birds generally have eyes located laterally on their skulls. The consequent high-acuity lateral vision helps with navigation and foraging behaviors during flight when visual attention is focused towards terrestrial features [28]. Their relatively small field of forward-facing binocular vision likely assists birds with more proximate tasks such as landing, perching, feeding, and provisioning of young [28, 29]. Taken together, this means that many birds flying in open air space are likely looking down and to the side rather than immediately in front of their bodies. Consequently, the visual physiology and behavior of birds flying at high altitudes may be insufficient to reliably detect structures in open air space when the structures are directly in line with a bird's flightpath [24, 30]. In addition, birds have evolved mostly in the absence of tall human-made structures. Hence, these structures present evolutionarily novel hazards for flying birds [30, 31].

How can we alter the visual attention of flying birds to increase the visibility of tall hazards? One way is to stimulate more than one sensory modality. Multi-modal signals increase the saliency and efficacy of communication signals by augmenting their detectability and decreasing the influence of unintended signals and background noise [32, 33]. For these reasons, multi-modal signals are prevalent in nature, particularly as warning signals [34, 35], and have also been used to mitigate other human-wildlife conflicts, such as in deterring birds from

conflict areas [36]. Multi-modal signals may be highly applicable for collision scenarios, where a greater detectability of warning signals is required. In particular, a combination of acoustic and visual cues can decrease collision risks of birds in captive flight trials [37]. Here, we extend those studies to perform a field test of a multimodal solution that combines an acoustically novel signal with the visual cues of tall communication towers in order to reduce collision risk for birds.

We designed the study to address three major objectives. First, we tested the efficacy of novel acoustic signals at reducing the risk of collisions with visually-conspicuous tall human-made structures in a field setting. We projected acoustic signals into the open airspace surrounding communication towers and evaluated the collision risk of flying birds in sound treatment and control (no broadcast sound) conditions. Second, we explored whether the frequency bandwidth characteristics of the acoustic signals influenced efficacy in reducing collision risk. Specifically, we compared the effects of a 4 to 6 kHz sound to those of a 6 to 8 kHz sound. Third, we developed new behavioral analyses to interpret collision risk and evaluate the efficacy of mitigation techniques. Much of the current data on collisions is estimated through direct observations of mortality. Researchers commonly conduct surveys of carcasses around the hazard, sometimes supplemented by anecdotal reporting from non-systematic survey methods [e.g. 38]. These methods of data collection likely lead to under-estimates of actual levels of collisions, often struggling to account for the influences of scavengers, delayed mortalities, or carcass persistence and detectability [39–41]. Furthermore, even non-fatal collisions may still negatively impact birds through, for example, increased energetic costs in avoiding hazards [42]. An important goal in evaluating the impacts of collision hazards on birds is to characterize avoidance, including through behavioral measures [43]. We analyzed the flight behaviors of birds to assess collision risk during sound treatment and control conditions, using videographic three-dimensional tracking techniques.

## Methods

### Ethics statement

Our experiments were conducted at communication tower sites in Virginia, USA. Authorization to access sites was obtained from entities with ownership or operational privileges. Permission to access towers was granted by the Delmarva Educational Association for the site in Townsend, VA, and by the Virginia State Police for the site in Eastville, VA. The field experimental protocol was approved by the William & Mary Institutional Animal Care and Use Committee (Protocol number: IACUC-2020-01-16-14047-jpswad). No additional public or private permits were required.

### Study area

Experiments were conducted between September and November 2019 at two communication tower sites on the Delmarva Peninsula in Virginia: a 107 m tall self-supported tower in Eastville operated by the Virginia State Police (37.299, -75.931; hereafter, *VSP* tower); and a 161 m tower supported by equally angled guy lines in Townsend operated by the Delmarva Educational Association (37.181, -75.963; hereafter, *DEA* tower). Both towers extend into open airspace above surrounding landscapes (a mix of forest and open multipurpose or agricultural land) with no buildings within 150 m (Fig 1). The Delmarva Peninsula is an important part of the Atlantic migratory flyway, with heavily used stopover habitat and consequently high avian diversity and abundance during the North American fall migration [44]. There are numerous communication towers along this peninsula as well as current and planned near- and offshore wind energy development, which will all pose collision risks to birds [45, 46].

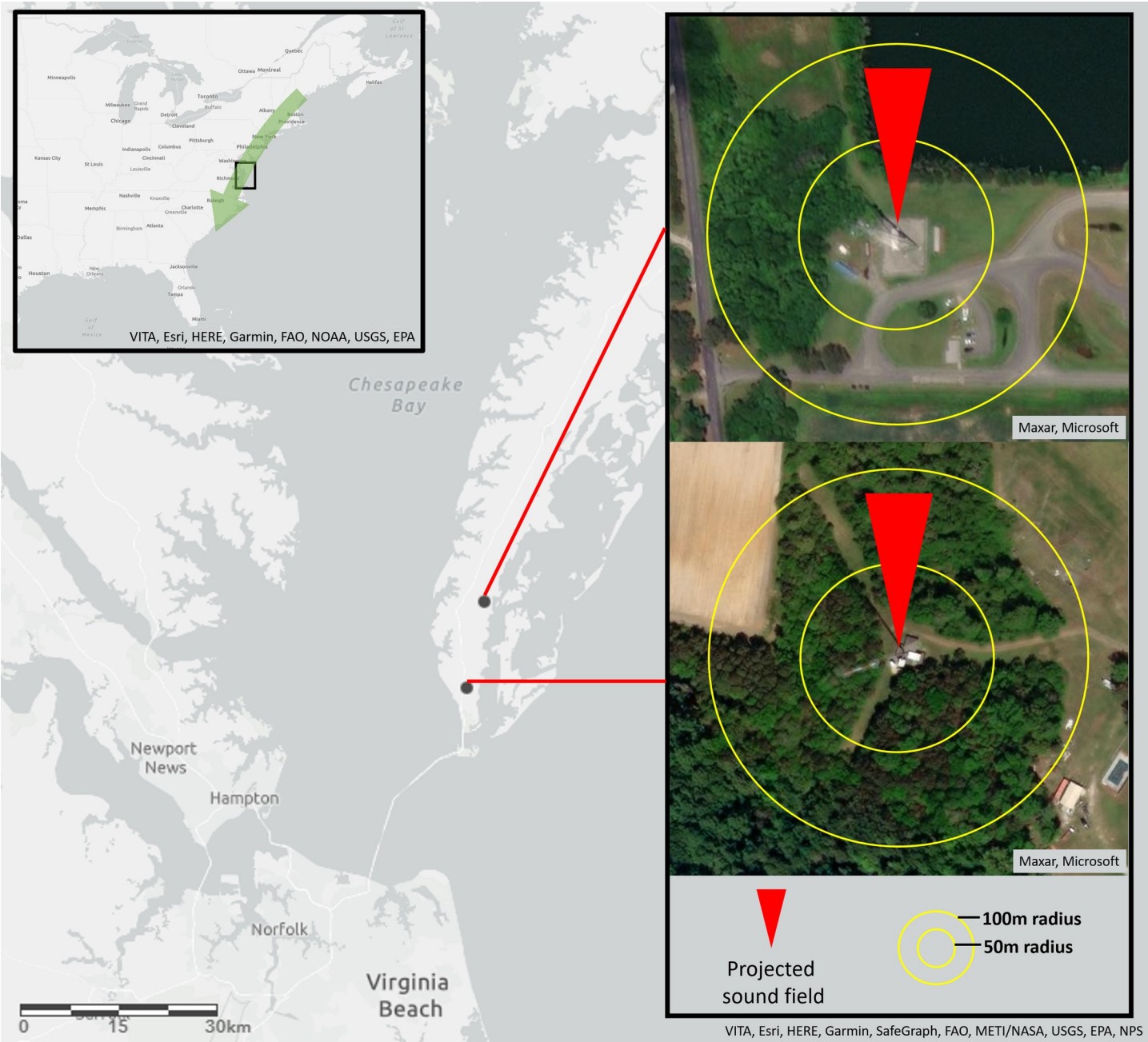

**Fig 1. Communication tower sites in Virginia.** Inset a) indicates location of Delmarva Peninsula. Inset b) indicates location of the *VSP* tower site in Eastville, VA. Inset c) indicates location of the *DEA* tower site in Townsend, VA. Sound fields were oriented northwards to match the general southward movements of migrating birds.

## Acoustic stimuli

We created two acoustic stimuli that were designed to lie within known avian auditory sensitivities and propagate with relatively low degradation through open-air environments with minimal masking by background noise [47, 48]. Specifically, we band-pass filtered white noise to generate 30 min duration sounds that lay between 4 to 6 kHz (S1 Audio) and 6 to 8 kHz (S2 Audio). Many songbirds have peak hearing sensitivity between 2 and 4 kHz and can clearly

hear sounds up to 8 kHz at 70 dBA SPL (see playback conditions below) [47, 49]. Many non-songbird species have peak sensitivities closer to 2 kHz and likely have more difficulty hearing sounds as high-pitched as 8 kHz at 70 dBA SPL [47, 50]. We generated these acoustic stimuli using PRAAT (version 6.0.46; [51]) and edited files in Audacity(R) (version 2.4.1; https://audacityteam.org/). During experiments, we alternated presentation of the noise stimuli with silent periods of equal 30 min duration (referred to as control periods). We projected stimuli at an intended intensity of >70 dBA SPL at 50 m, by adjusting speaker output to 100 dBA SPL at 3 m from the speaker, as measured by a handheld sound pressure level meter (Extech 407730 with dBA weighting and slow integration time), and assuming an average attenuation rate of -6 dB per doubling of distance through open-air (i.e. spherical spreading of sound).

## General experimental procedure

Experiments were conducted using a fully factorial design between site and stimulus type: Each site (*DEA* and *VSP*) experienced each stimulus type (4–6 kHz and 6–8 kHz) on three separate day replicates. Thus, data were collected on six separate days over the course of the season. Experiments were run on pairs of consecutive days, with one stimulus type being used at both sites on the first day and the second stimulus type being used at both sites on the subsequent day. The order of stimulus type use in a pair of days was continually counterbalanced through the course of the study.

We conducted our field experiments between 05:00 and 11:00 as this is a period of high collision incidence [16, 52]. We projected sound fields from the base of each tower using a highly directional speaker (LRAD 100X) positioned at an angle of 45˚ from the horizon. The sound field was oriented northwards, to match the general southward movements of migrating birds (Fig 1). We alternated 30 min periods of sound stimuli projection with 30 min periods of silent control. Treatment and control periods were separated by 15 min buffer intervals of no sound and no data collection, intended to minimize any spill-over effect between treatment periods. We recorded videos during treatment and control periods with two GoPro cameras (Hero7 Black) positioned 1.2 m either side of the speaker at the base of the tower. Both cameras were angled upwards at an angle of 45˚ from the horizon and angled inwards toward a common focal point 3 m in front of the speaker, to triangulate on the area of interest around the tower encompassing the sound field.

For each day of data collection and each tower location, we extracted average daily temperature (˚C) and precipitation (mm) from the PRISM Climate Group gridded dataset (Oregon State University, http://www.prism.oregonstate.edu) and scored cloud cover (oktas) from aggregate video data. Experiments were not conducted on days with heavy precipitation or average wind speeds above 2 m/s, in an effort to minimize variation in abiotic variables between sampling times.

## Flight analyses

We visually inspected all video recordings to classify the general size of birds (*small*: songbird-sized; or *large*: medium-size birds such as *Corvidae* and larger such as *Buteo jamaicensis*), the group size (*single* or *flock* of any size), and the number of bird flights around each tower (less than 500 m) during each treatment period of the experiment. Further, we identified flights for analysis where the bird(s) flew centrally through the fields of view of video cameras, below the maximum height of the towers but at least 5 m above the ground, and where birds did not perch on surrounding vegetation or on the towers or guy ropes themselves. This procedure allowed us to identify 'at risk' flights where birds were most likely to have not recently interacted with the tower (i.e., they did not perch) and were at most risk of collision with the tower.

We analyzed the 'at risk' flights to further describe flight behaviors. We used stereographic video data (at 30 frames per second) to recreate the three-dimensional (3D) coordinates of bird flights surrounding towers, using methods available in the open-source Argus packages implemented in Python 3.6.2 [53]. In order to maximize coverage of the airspace around towers, we used a wide angle (focal length: 15 mm) setting on the cameras. We synchronized the cameras with the playback of conspicuous audio tones. We video recorded a drone flying through the focal airspace carrying a 1 m calibration wand (a wooden dowel with painted polystyrene balls on either end) at the beginning of each experimental day in order to adequately calibrate the active airspace in later videos. We calculated intrinsic camera parameters, including lens distortion due to the wide-angle mode, by recording a dot calibration pattern and building an omnidirectional camera profile in Python 3.6.2. To recreate 3D flight paths of all birds, we calibrated the cameras and airspace using a wand-based, direct linear transformation (DLT) technique with sparse-bundle adjustment (SBA), implemented in the Argus packages in Python 3.6.2 [53]. In both camera views, we manually digitized the centroid of each bird for every frame during a flight to determine the 3D positions with respect to time. To reduce digitization noise, these raw data were smoothed using a quintic spline function. The spline error tolerances were weighted by error variances extracted from the 3D reconstruction uncertainty at every data point. These calibrations produced root mean square reprojection errors of between 0.5 and 0.8 pixels for the two cameras. Variation in the reconstructed wand length, expressed as the ratio of the standard deviation divided by the mean and multiplied by 100, ranged from 5.33 to 13.6. This related to standard deviation of between 0.032 and 0.082 m of the 1 m wand length, within a filming volume of approximately 1 500 000 m$^3$.

Three points from right-angled fence posts were used to define references axes at both tower sites. These axes were then transformed so that the origin was at the base of the towers, the x axis was north-south, and the z axis was vertical. This allowed flight paths to be measured on a global reference system related to the towers.

We derived a set of three instantaneous flight metrics from smoothed 3D positional data to characterize the flight behavior of birds. Horizontal distance ($d$) from the tower was calculated for every frame of a bird's flightpath:

$$d = \sqrt{x^2 + y^2}$$

where x and y are the coordinates of a bird in the horizontal plane, with the tower as the origin, for a given frame. Velocity was estimated as the first derivative of position with respect to time from the quintic spline polynomial. Absolute velocity ($v$) was calculated from the derived velocities for each axis of a bird's flightpath:

$$v = \sqrt{V_x^{\,2} + V_y^{\,2} + V_z^{\,2}}$$

where $V_x$, $V_y$ and $V_z$ are the estimated velocities in each 3D plane, for a given frame. Horizontal angle between a bird's momentary flight trajectory and the tower ($\Theta_{tower}$) was calculated for every frame of a bird's flightpath:

$$\theta_{tower} = \cos^{-1}\left(\frac{(\vec{A} \cdot \vec{O})}{\|\vec{A}\|\|\vec{O}\|}\right)$$

where $\vec{A}$ is a vector between consecutive 3D positions in a bird's flightpath and $\vec{O}$ is the vector between the same starting position and the tower.

Each metric was then represented for every flight by taking the median measure of all the frame-by-frame estimates. A secondary set of measures were derived to capture changes in

**Table 1. Summary of all flight metrics and calculations.**

| Metric | Formula | Overall flight | Overall flight interpretation | Within-flight change | Within-flight interpretation |
|---|---|---|---|---|---|
| Horizontal distance from tower | $d = \sqrt{x^2 + y^2}$ | $MED_d =$ median($d$) | Larger median values of $d$ mean greater distances from towers, conferring less collision risk | $\Delta MED_d = MED_d(later)$– $MED_d(earlier)$ | Negative values of $\Delta MED_d$ mean flights approach towers. Smaller negative values mean flights approach towers less, conferring less collision risk |
| Absolute velocity | $v = \sqrt{V_x{}^2 + V_y{}^2 + V_z{}^2}$ | $MED_v =$ median($v$) | Smaller median values of $v$ mean slower flight velocities, conferring less collision risk | $\Delta MED_v = MED_v(later)$– $MED_v(earlier)$ | Positive values of $\Delta MED_v$ mean acceleration of flights. Smaller positive values mean less acceleration, conferring less collision risk |
| Angle of displacement from tower | $\theta_{tower} = \cos^{-1}\left( \dfrac{(\vec{A} \cdot \vec{O})}{\|\vec{A}\| \|\vec{O}\|} \right)$ | $MED_{\Theta tower} =$ median ($\Theta_{tower}$) | Larger median values of $\Theta_{tower}$ mean greater angles of displacement, conferring less collision risk | $\Delta MED_{\Theta tower} =$ $MED_{\Theta tower}(later)$– $MED_{\Theta tower}(earlier)$ | Positive values of $\Delta MED_{\Theta tower}$ mean increased angles of displacement from towers. Larger positive values mean greater increased angles, conferring less collision risk |

$d$ is the horizontal distance from towers; $v$ is the absolute velocity; $\Theta_{tower}$ is the horizontal angle between the momentary flight trajectory and a reference trajectory to towers. Interpretations describe potential metric outcomes and their predicted influence on collision risk.

these flight behavior metrics through the course of a bird's flight. To achieve this, a bird's flight path was divided equally into its temporally earlier and later halves, capturing how the bird responds to the increasing loudness of the sound cue and increasing proximity to the tower. Each of the three metrics ($d$, $v$, $\Theta_{tower}$) was summarized for both halves of the flight. Finally, the change in the median measure from the earlier half of the flight to the later half (e.g. change in median distance ($\Delta d$) = median $d_{later\ half}$–median $d_{earlier\ half}$) of the flight were calculated for each metric. Table 1 summarizes all flight metrics and calculations. An example flight and its derived metrics are presented in S1 Fig. Calculations were executed using R (R Core Team, 2019; [54]) statistical software.

## Statistical analyses

We analyzed flight behavior metrics using multiple linear regression analysis implemented in R (R Core Team, 2019). Specifically, we modeled the six flight metrics summarized in Table 1. All outcome metrics met the normality assumption of linear regression and were modeled using a normal (Gaussian) error structure. The overall flight metrics generated data bound by zero but, under a normal error structure, produced almost perfectly linear relationships with normally distributed deviance residuals and did not predict non-sense values. Treatment condition (*4–6 kHz*, *6–8 kHz*, *control*), date (6 levels), tower site (*DEA*, *VSP*), bird size (*small*, *large*), and bird group size (*single*, *flock*) were treated as categorical fixed factors. A set of candidate models was built from a priori hypotheses and from explorations of non-linearity between predictors and response variables and of collinearity between predictor variables (S1 Table). The same initial set of candidate models was used for all outcome variable analyses.

Models were evaluated using Akaike's information criteria (AIC) with small sample bias adjustment, AICc. First, AICc weight was used to rank model suitability and top-performing models, with AICc weights adding up to 95% of total AICc weight, were preserved. Then, more complex models in a nested set of models which performed worse than their simpler forms were excluded [55]. Outcome variable predictions were generated for each model from these final sets and then averaged by AICc weight to produce model-averaged estimates of each of the 6 outcome variables, with standard errors [56]. Top-performing models were also used to compute model-averaged predictor variable parameter estimates and 95% confidence intervals, using subset averaging [56]. Given some of the challenges in interpreting model

averaged estimates of predictor parameters, due to potential multi-collinearity among predictors [57], inferences were made partly from model averaged outcome predictions and their standard errors. Specifically, the marginal effects of predictors of interest on the model averaged outcomes were explored by summarizing outcomes by predictor groups using the mean plus or minus the standard error of the mean. When multi-collinearity among predictor variables was not a concern, the statistical significance of model averaged predictor parameters was explored [57].

## Results

### Bird activity around towers

We collected data on 6 sampling days from two tower locations, for three hours each day, generating 9 hours of video footage at each tower site for each sound frequency. We inspected the rate of detected bird flights (number of flights per minute) around each tower during each treatment period of the experiment. Overall, we logged 1585 interactions between towers and birds. There was a 16.2% lower mean rate of detections during 4–6 kHz sound treatment periods (mean = 1.29; SE ± 0.34) compared to control periods (mean = 1.54; SE ± 0.31), and an 11.7% lower mean rate of detections during 6–8 kHz sound treatment periods (mean = 1.36; SE ± 0.5) compared to control periods (Fig 2). Hence, there is a correlation between treatment conditions and bird activity levels and it is possible that the sound stimuli deterred some birds from entering the general vicinity of the towers.

### Flight behavior metrics of "at risk" flights

Of the 1585 total interactions between birds and towers, 145 (9.1%) were deemed "at risk" flights, for subsequent behavioral analysis. The mean "at risk" interaction rate (birds per minute) was 0.12 (SE ± 0.02) during control periods, 0.13 (SE ± 0.03) during 4–6 kHz treatment periods, and 0.14 (SE ± 0.05) during 6–8 kHz treatment periods. Despite overall differences in general bird activity between tower sites, the rate of "at risk" interactions did not vary substantially between treatment condition periods within tower sites.

**Distance from towers.** The final set of selected models fit to the overall distance outcome variable, ordered by delta AICc value and derived model weight, are described in S2 Table. The 4–6 kHz treatment condition significantly influenced overall distance from towers compared to the control condition, with model-averaged 95% confidence intervals not overlapping zero (Table 2). The effect size of this parameter estimate was estimated as a 5.18 m greater mean overall distance from towers during 4–6 kHz treatment conditions (Table 2; Fig 3A). The effect size of the 6–8 kHz treatment estimated flights that were 1.19 m closer to towers, compared to control conditions, though this effect was not statistically supported according to interpretation of 95% confidence intervals (Table 2; Fig 3A). Treatment condition, tower site and sampling date dominated top-performing models (S2 Table), thereby heavily influencing model averaged estimates. In particular, we note that sampling date influenced the proximity that birds reached to towers (Table 2).

The final set of selected models fit to the within-flight change in distance outcome variable, ordered by delta AICc value and derived model weight, are described in S3 Table. Neither the 4–6 kHz nor the 6–8 kHz treatment conditions significantly influenced the within-flight change in distance, with 95% confidence intervals overlapping zero (Table 3) and model averaged marginal predictions with overlapping standard error bars (Fig 3B). Though not statistically supported, the 4–6 kHz and 6–8 kHz treatments were associated with a 2.62 m (or 2.9 m from marginal predictions) and 4.67 m (or 4.0 m from marginal predictions), respectively, increase in the change in distance compared to control conditions (Table 3; Fig 3B). An

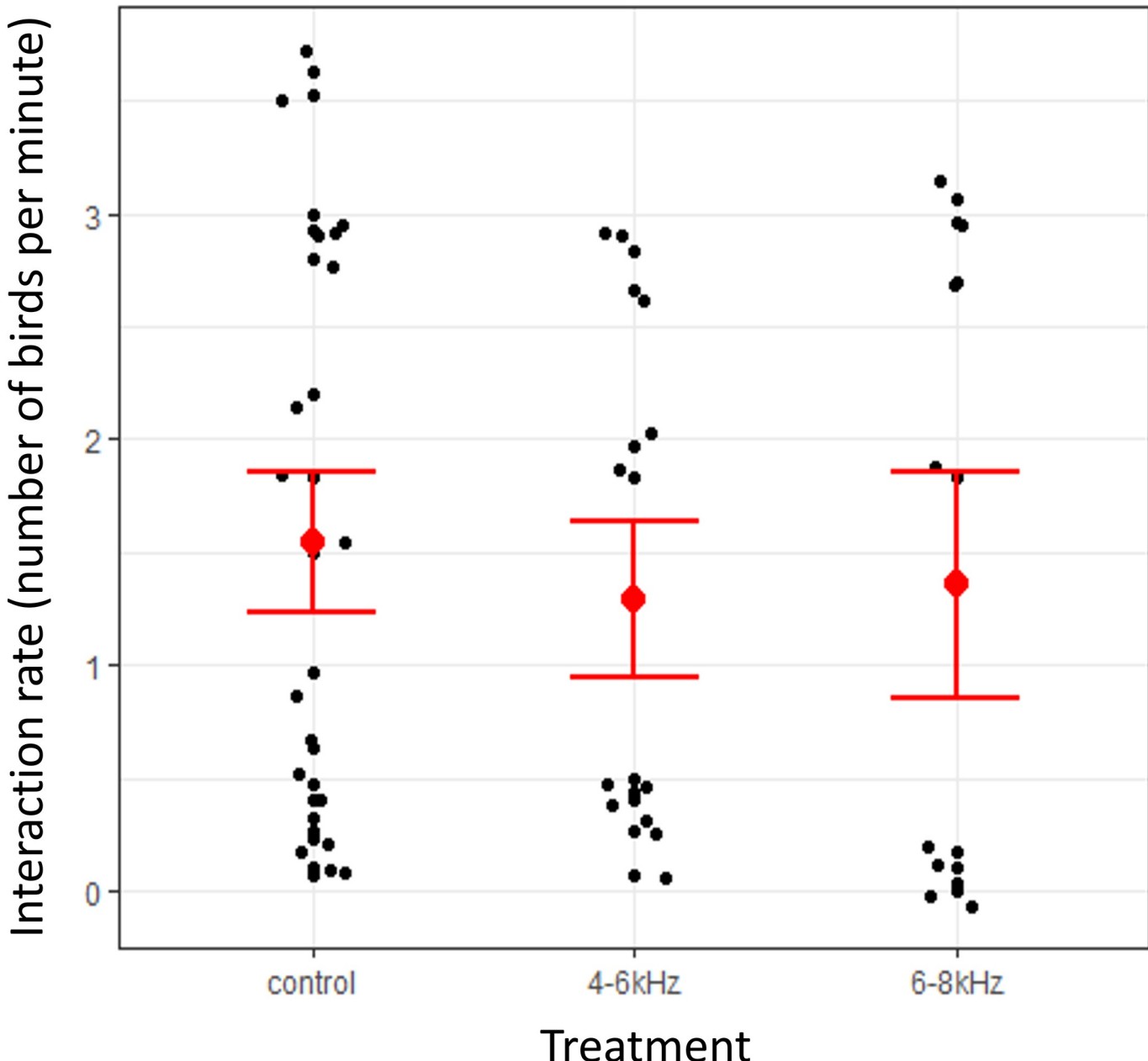

**Fig 2. General bird activity around towers by treatment condition.** The overall rate of detections by treatment condition. Solid black dots represent detection rates per 30-minute sampling period. Solid red dots and error bars represent mean detection rate ± standard error of the mean.

increase in change in distance indicates that bird flights remained further away from towers during treatment conditions compared to control conditions (Fig 3B). Treatment condition, bird size, sampling date and tower site dominated top-performing models (S3 Table), thereby heavily influencing model averaged estimates. The interaction term between VSP site and 6–8 kHz treatment condition significantly influenced change in distance, with the 95% confidence interval not overlapping zero (Table 3). The marginal predictions of change in distance by treatment condition and tower site are plotted in S2 Fig. The interaction term between small bird size and 6–8 kHz treatment condition significantly influenced change in distance, with

**Table 2. Overall distance model-averaged parameter estimates.**

| Parameter | beta coefficient | Lower 95% CI | Upper 95% CI |
|---|---|---|---|
| (Intercept) | 21.9 | 16.4 | 27.4 |
| Treatment 4-6kHz | 5.18 | 0.672 | 9.68 |
| Treatment 6-8kHz | -1.19 | -5.76 | 3.37 |
| Site VSP | 2.98 | -0.836 | 6.80 |
| Date 100119 | 4.50 | -0.867 | 9.86 |
| Date 110319 | 12.2 | 5.54 | 18.8 |
| Date 110419 | 5.96 | 0.598 | 11.3 |
| Date 092419 | -1.92 | -8.50 | 4.66 |
| Date 093019 | 7.45 | 0.076 | 14.8 |

Subset of all model-averaged parameter estimates based on a-priori interest and parameter importance. Treatment refers to the effects of the two sound ranges. Site refers to the VSP tower location. Date refers to the date (month-day-year) of observations. Parameter estimates with 95% confidence intervals not overlapping zero are highlighted in gray.

the 95% confidence intervals not overlapping zero (Table 3). The marginal predictions of change in distance by treatment condition and bird size are plotted in S3 Fig.

**Flight velocity.** The final set of selected models fit to the overall velocity outcome variable, ordered by delta AICc value and derived model weight, are described in S4 Table. The 4–6 kHz and 6–8 kHz treatment conditions significantly influenced overall velocity compared to the control condition, with 95% confidence intervals not overlapping zero (Table 4). The effect size of the 4–6 kHz parameter was related to an estimated 1.47 m/s lower mean velocity during

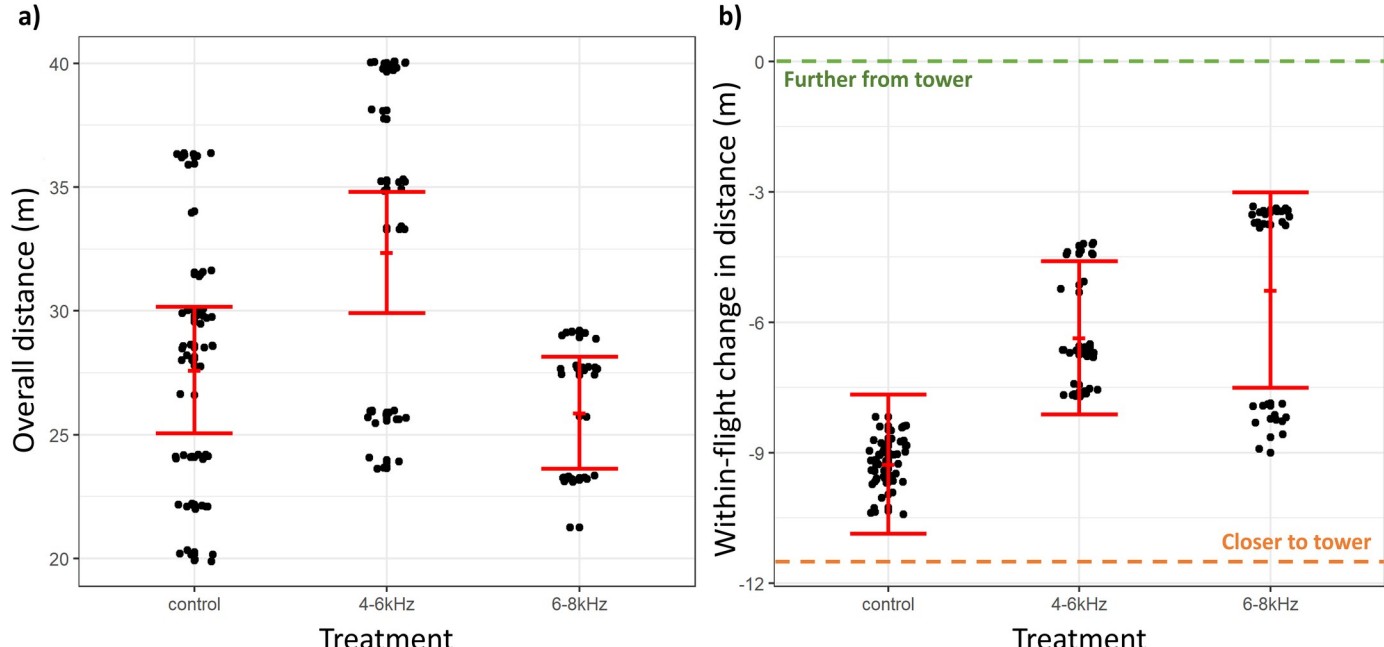

**Fig 3. Distance from towers by treatment condition.** Panel a) shows the overall distance from towers by treatment condition. Panel b) shows the within-flight change in distance from towers by treatment condition. Solid red dots and error bars represent model-averaged mean of outcome variable ± standard error of the mean. In panel b), the green dotted line indicates a level of change in distance where flights remain further away from towers and the orange dotted line indicates a level of change in distance where flights draw closer to towers.

**Table 3. Within-flight change in distance model-averaged parameter estimates.**

| Parameter | beta coefficient | Lower 95% CI | Upper 95% CI |
|---|---|---|---|
| (Intercept) | -9.22 | -11.9 | -6.55 |
| Treatment 4-6kHz | 2.62 | -1.60 | 6.84 |
| Treatment 6-8kHz | 4.67 | -4.03 | 13.4 |
| Site VSP | -1.87 | -5.47 | 1.73 |
| Site VSP: Treatment 4-6kHz | 1.20 | -4.54 | 6.94 |
| SiteVSP: Treatment 6-8kHz | 11.7 | 2.00 | 21.3 |
| Bird size small | 0.573 | -2.77 | 3.92 |
| Bird size small: Treatment 4-6kHz | 3.00 | -2.43 | 8.42 |
| Bird size small: Treatment 6-8kHz | -7.55 | -13.2 | -1.88 |
| Date 100119 | 1.71 | -2.74 | 6.16 |
| Date 110319 | -4.14 | -9.87 | 1.59 |
| Date 110419 | -0.778 | -5.43 | 3.88 |
| Date 092419 | 2.16 | -2.35 | 6.67 |
| Date 093019 | 3.15 | -2.50 | 8.81 |

Subset of all model-averaged parameter estimates based on a-priori interest and parameter importance. Treatment refers to the effects of the two sound ranges. Site refers to the VSP tower location. Bird size refers to the small bird size class. Date refers to the date (month-day-year) of observations. Parameter estimates with 95% confidence intervals not overlapping zero are highlighted in gray.

4–6 kHz treatment conditions compared to control conditions (Table 4; Fig 4A). The effect size of the 6–8 kHz parameter was related to an estimated 1.85 m/s greater mean velocity during 6–8 kHz treatment conditions compared to control conditions (Table 4; Fig 4A). Treatment condition, tower site and sampling date dominated top-performing models (S4 Table), thereby heavily influencing model averaged estimates. In particular, sampling date and bird size noticeably influenced flight velocity, with parameter estimates only marginally overlapping zero (Table 4).

**Table 4. Overall velocity model-averaged parameter estimates.**

| Parameter | beta coefficient | Lower 95% CI | Upper 95% CI |
|---|---|---|---|
| (Intercept) | 7.09 | 4.96 | 9.22 |
| Treatment 4-6kHz | -1.47 | -2.69 | -0.251 |
| Treatment 6-8kHz | 1.85 | 0.622 | 3.09 |
| Site VSP | -0.312 | -2.60 | 1.98 |
| Date 100119 | -1.03 | -4.78 | 2.72 |
| Date 110319 | 3.16 | -0.298 | 6.62 |
| Date 110419 | 2.70 | -0.852 | 6.26 |
| Date 092419 | 0.945 | -1.62 | 3.51 |
| Date 093019 | -1.39 | -4.69 | 1.91 |
| Bird group single | -0.945 | -2.06 | 0.173 |
| Bird size small | 0.896 | -0.056 | 1.85 |

Subset of all model-averaged parameter estimates based on a-priori interest and parameter importance. Treatment refers to the effects of the two sound ranges. Site refers to the VSP tower location. Date refers to the date (month-day-year) of observations. Bird group refers to the single bird category. Bird size refers to the small bird size class. Parameter estimates with 95% confidence intervals not overlapping zero are highlighted in gray.

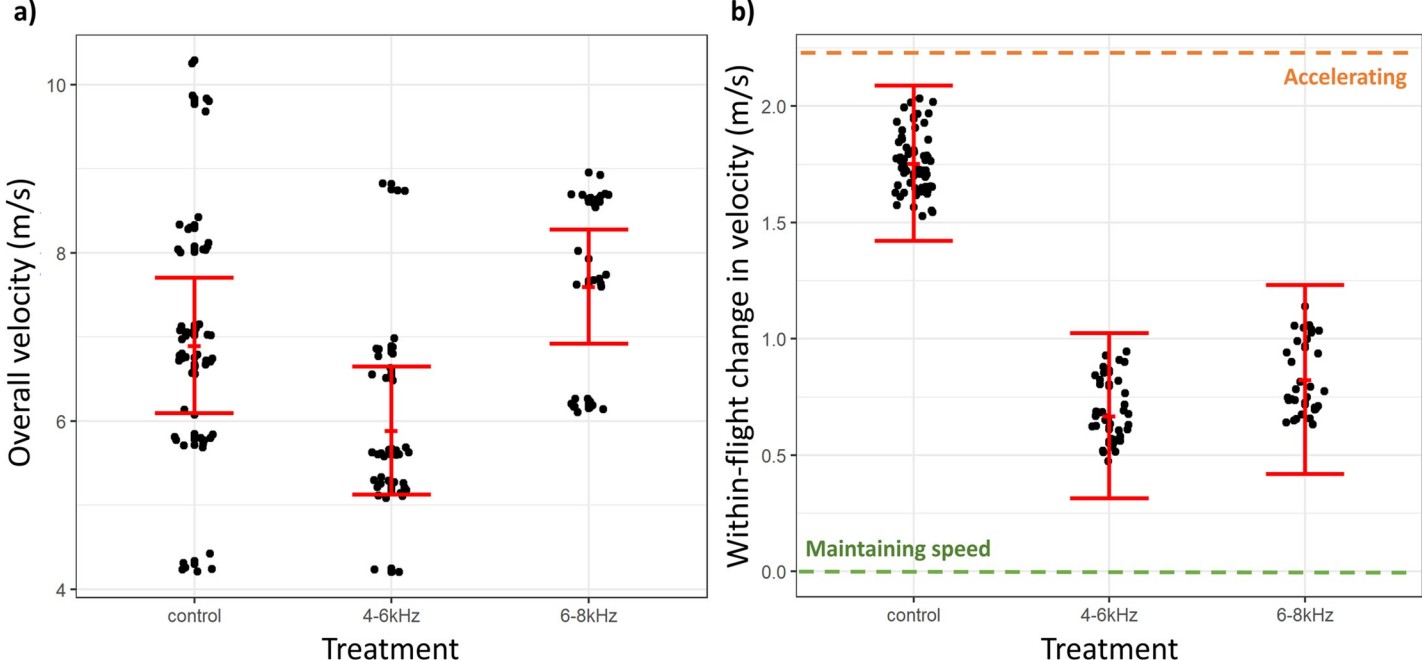

**Fig 4. Flight velocity by treatment condition.** Panel a) shows the overall flight velocity by treatment condition. Panel b) shows the within-flight change in flight velocity by treatment condition. Solid red dots and error bars represent model-averaged mean of outcome variable ± standard error of the mean. In panel b), the green dotted line indicates the level of change in velocity representing a maintaining of flight speed and the orange dotted line indicates the level of change in velocity representing an acceleration in flight speed.

The final set of selected models fit to the within-flight change in velocity outcome variable, ordered by delta AICc value and derived model weight, are described in S5 Table. The 4–6 kHz treatment condition significantly influenced change in velocity, with the 95% confidence interval not overlapping zero (Table 5) and model averaged marginal predictions with non-overlapping standard error bars (Fig 4B). The effect size of the 4–6 kHz parameter was related to an estimated 1.01 m/s (or 1.08 m/s from marginal predictions) decrease in the change in velocity compared to control conditions (Table 5; Fig 4B). A smaller change in velocity indicates that birds increased their speed less through the course of flights during 4–6 kHz conditions compared to control conditions. The 6–8 kHz treatment condition parameter had 95% confidence intervals overlapping zero (Table 5) but had non-overlapping standard error bars

**Table 5. Within-flight change in velocity model-averaged parameter estimates.**

| Parameter | beta coefficient | Lower 95% CI | Upper 95% CI |
| --- | --- | --- | --- |
| (Intercept) | 1.56 | 0.813 | 2.30 |
| Treatment 4-6kHz | -1.01 | -1.90 | -0.112 |
| Treatment 6-8kHz | -0.671 | -2.62 | 1.28 |
| Site VSP | 0.407 | -0.552 | 1.37 |
| Site VSP: Treatment 4-6kHz | -0.764 | -2.42 | 0.892 |
| Site VSP: Treatment 6-8kHz | -2.93 | -5.73 | -0.131 |
| Bird size small | 0.550 | -0.132 | 1.23 |

Subset of all model-averaged parameter estimates based on a-priori interest and parameter importance. Treatment refers to the effects of the two sound ranges. Site refers to the VSP tower location. Bird size refers to the small bird size class. Parameter estimates with 95% confidence intervals not overlapping zero are highlighted in gray.

between 6–8 kHz and control conditions in model averaged marginal predictions (Fig 4B). The 6–8 kHz treatment condition parameter was related to an estimated 0.67 m/s (or 0.93 m/s from marginal predictions) decrease in the mean outcome of the change in velocity, compared to control conditions (Table 5; Fig 4B). This too indicates that birds increased speed less through the course of flights during 6–8 kHz conditions compared to control conditions. Treatment condition, bird size, and tower site dominated top-performing models (S5 Table), thereby heavily influencing model averaged estimates. The interaction term between VSP site and 6–8 kHz treatment condition significantly influenced change in velocity, with the 95% confidence interval not overlapping zero (Table 5). The marginal predictions of change in velocity by treatment condition and tower site are plotted in S4 Fig.

**Angle of displacement.** The final set of selected models fit to the overall angle of displacement outcome variable, ordered by delta AICc value and derived model weight, are described in S6 Table. The 6–8 kHz treatment condition significantly influenced the overall angle of displacement compared to the control condition, with 95% confidence intervals not overlapping zero (Table 6) and model averaged marginal predictions with non-overlapping standard error bars (Fig 5A). The effect size of the 6–8 kHz parameter was related to an estimated 20.0˚ (or 14.1˚ from marginal predictions) increase in the mean angle of displacement, compared to control conditions (Table 6; Fig 5A). The 4–6 kHz treatment condition marginally influenced the mean angle of displacement, compared to control condition, based on 95% confidence intervals only just overlapping zero (Table 6) and model averaged marginal predictions with non-overlapping standard error bars (Fig 5A). The effect size of the 4–6 kHz parameter was related to an estimated 10.5˚ (or 13.5˚ from marginal predictions) increase in the mean angle of displacement, compared to control conditions (Table 6; Fig 5A). Treatment condition and bird size dominated top-performing models, thereby heavily influencing model averaged estimate (S6 Table).

The final set of selected models fit to the within-flight change in angle of displacement outcome variable, ordered by delta AICc value and derived model weight, are described in S7 Table. Neither the 4–6 kHz nor the 6–8 kHz treatment conditions significantly influenced change in angle of displacement, based on 95% confidence intervals overlapping zero (Table 7) and model averaged marginal predictions with overlapping standard error bars (Fig 5B). Though not statistically supported, the 4–6 kHz and 6–8 kHz treatments were associated with a 0.48˚ and 12.7˚, respectively, increase in the change in angle of displacement compared to control conditions (Table 7; Fig 5B). An increase in change in angle of displacement indicates that bird flights angled further away from towers during treatment conditions compared to

**Table 6. Overall angle of displacement model-averaged parameter estimates.**

| Parameter | beta coefficient | Lower 95% CI | Upper 95% CI |
|---|---|---|---|
| (Intercept) | 56.7 | 49.2 | 64.2 |
| Treatment 4-6kHz | 10.5 | -0.290 | 21.3 |
| Treatment 6-8kHz | 20.0 | 6.70 | 33.3 |
| Site VSP | -0.723 | -10.3 | 8.83 |
| Bird group single | -3.72 | -13.4 | 6.01 |
| Bird size small | 0.135 | -11.4 | 11.7 |

Subset of all model-averaged parameter estimates based on a-priori interest and parameter importance. Treatment refers to the effects of the two sound ranges. Site refers to the VSP tower location. Bird size refers to the small bird size class. Bird group refers to the single bird category. Parameter estimates with 95% confidence intervals not overlapping zero are highlighted in gray.

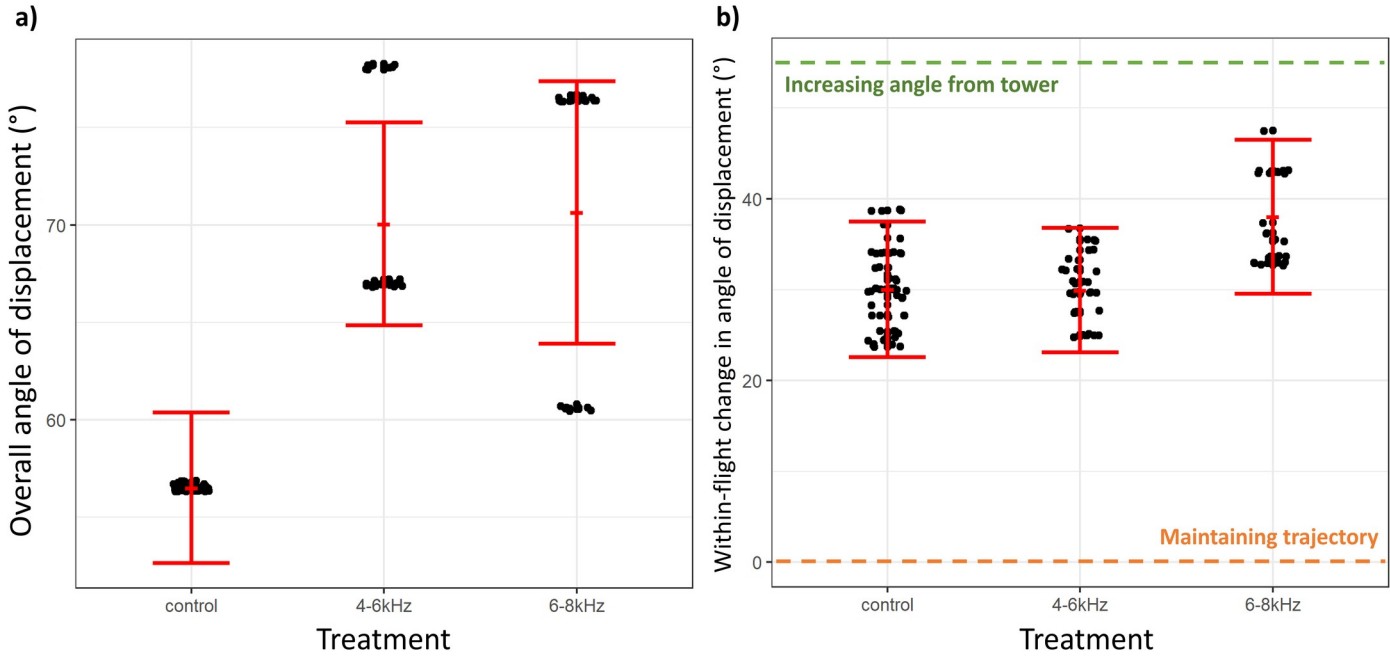

**Fig 5. Angle of displacement by treatment condition.** Panel a) shows the overall angle of displacement by treatment condition. Panel b) shows the within-flight change in angle of displacement by treatment condition. Solid red dots and error bars represent model-averaged mean of outcome variable ± standard error of the mean. In panel b), the green dotted line indicates an increasing angle of displacement from towers and the orange dotted line indicates a maintenance of trajectory towards towers.

control conditions (Fig 5B). Treatment condition, tower site and sampling date dominated top-performing models (S7 Table), thereby heavily influencing model averaged estimates. In particular, we note that sampling date influenced the angle of displacement from towers (Table 7).

## Discussion

Bird activity rates around communication towers were 16.2% lower during 4–6 kHz sound treatment conditions compared to control conditions and 11.7% lower during 6–8 kHz sound

**Table 7. Within-flight change in angle of displacement model-averaged parameter estimates.**

| Parameter | beta coefficient | Lower 95% CI | Upper 95% CI |
|---|---|---|---|
| (Intercept) | 41.3 | 19.7 | 62.9 |
| Treatment 4-6kHz | -0.480 | -12.9 | 12.0 |
| Treatment 6-8kHz | 12.7 | -0.425 | 25.8 |
| Site VSP | -7.88 | -19.7 | 3.89 |
| Date 100119 | -20.6 | -37.0 | -4.29 |
| Date 110319 | -8.13 | -26.4 | 10.2 |
| Date 110419 | -19.1 | -35.8 | -2.42 |
| Date 092419 | -17.6 | -35.4 | 0.148 |
| Date 093019 | -5.70 | -26.2 | 14.8 |

Subset of all model-averaged parameter estimates based on a-priori interest and parameter importance. Treatment refers to the effects of the two sound ranges. Site refers to the VSP tower location. Date refers to the date (month-day-year) of observations. Parameter estimates with 95% confidence intervals not overlapping zero are highlighted in gray.

treatment conditions compared to control conditions (Fig 2). Although these differences in activity rates did not translate into differences in the rates of "at-risk" interactions with towers, it is possible that the acoustic treatments deterred birds from the general vicinity (less than 500m) of the towers, thereby reducing the overall risk of collisions.

In addition to potentially deterring birds from entering the air space around towers, the acoustic treatments also altered flight behaviors of birds that entered the area within 100 m of each tower, further reducing the overall risk of collision. Specifically, the 4–6 kHz sound resulted in birds flying approximately 1.5 m/s or 14.6% slower and 5 m or 17.4% further away from the towers, on a heading that was an additional 10˚ or 23.9% away from the tower, relative to flights in control conditions (Figs 3–5). While it is difficult to translate these changes in flight behavior to a precise metric of collision, a bird is clearly at substantially reduced risk of collision when flying slower and further away from an approximately 5 m-wide tower. Furthermore, the acoustic lighthouse treatment gives the flying birds more time and space to react to the collision hazard, further reducing the risk of collision [58].

Within-individual changes in flight behavior, as the birds approached towers, further support the conclusion that the 4–6 kHz acoustic lighthouse treatment reduced collision risk. When exposed to this sound treatment, birds accelerated less and remained an additional 2.5 to 3.0 m further away from towers, relative to typical flights from birds in control conditions (Figs 3 and 4). Interestingly, the within-individual change in direction of heading of birds exposed to 4–6 kHz did not vary substantially from the change of heading of birds in control conditions, indicating that birds exposed to the 4–6 kHz sound treatment altered the direction of flight early in their flight and at larger distance from the tower. These patterns help us to understand how birds are reducing their collision risk during the acoustic treatments. Relative to control conditions, birds observed during treatment conditions appear to be slowing their flight down and making an early adjustment to their direction of flight so that they maintain a greater distance from the hazard as they pass through the area. These observations are consistent with our prediction that birds will pay greater attention to the hazard when the acoustic lighthouse is deployed.

When exposed to the higher frequency and presumably less audible 6–8 kHz sound, birds flew faster at greater displacement angles from towers but at similar distances from towers, compared to control conditions. Collectively, these observations do not uniformly meet our predictions and offer less evidence that the 6–8 kHz sound reduced the risk of in-flight collision. Examining the within-individual changes in flight metrics during the 6–8 kHz treatment sheds more light on how this sound is influencing collision risk. Though not statistically significant, flights during 6–8 kHz treatment conditions had a smaller within-flight decrease in distance from towers during their flight trajectories, compared to within-flight changes for birds in control conditions (Fig 3). This means that birds in this sound treatment slightly altered the latter part of their flight to avoid the tower by a greater distance—but only in the air space close to the tower. Birds in the 6–8 kHz treatment also accelerated less during the latter part of their flight, compared to control conditions (Fig 4), and somewhat altered the heading of the latter section of their flight further away from towers compared to adjustments made by birds in control conditions (Fig 5). Interpreting these within-individual modifications in flight in the context of the assessments of among individual flight patterns is consistent with birds taking delayed evasive action in the 6–8 kHz treatment compared with the 4–6 kHz treatment.

A delayed and less robust response to the 6–8 kHz sound compared with the 4–6 kHz sound supports our prediction that the 4–6 kHz sound is more audible and will be more effective in altering the birds' attention and perception of risk during flight. Reviews of avian auditory sensitivity indicate that most bird species have more sensitive hearing in the 4–6 kHz compared with the 6–8 kHz range [47, 49, 50]. We tested the effects of the 6–8 kHz sound as

there should be less interference and possibly less auditory masking with this sound because it is clearly further frequency-shifted from the lower frequency background noise of the environments in which we performed our tests [48]. However, we provide evidence that intermediate frequency signals, 4–6 kHz specifically, which more closely target the known peak auditory sensitivities of most birds, propagate with suitable detectability through open airspace. In further studies, we are comparing the utility of even lower frequency signals, such as a 2–4 kHz sound, which would target the most sensitive portions of the general avian auditory sensitivity spectrum [47] and would propagate generally further with less attenuation through open air [48] than the 4–6 and 6–8 kHz sounds, but may be more prone to frequency-dependent masking from background ambient noise for birds in flight. We are also currently exploring the role of frequency modulation in increasing signal detectability and collision avoidance.

Other than demonstrating the effectiveness of the 4–6 kHz sound in reducing collision risk, our study helps to demonstrate the value of behavioral methods in designing and assessing mitigation technology. By analyzing videos of flight behaviors, we quickly generated a robust sample size enabling statistically-supported assessment of the acoustic lighthouse concept. We could also discern degrees of avoidance of the hazard, which gives more quantitative power to the assessment of mitigation effectiveness. A traditional assessment of mortality at the sites would have taken many years instead of a couple of months, as observations of carcasses around towers are not common. Hence, the approach we used in this study can help assess mitigation technology more rapidly, saving time, money, and potentially leading to earlier implementation that would then improve conservation outcomes. We are not advocating for the cessation of mortality surveys. They should run in parallel with behavioral methods to help contextualize behavioral data.

Our novel application of behavioral methods also offers a more nuanced perspective on collision risk than mortality surveys. Not all collisions are fatal, and hazardous structures may present non-lethal challenges to birds even when collisions are avoided [43]. For example, avoiding hazards may increase energetic expenditure, which could prove costly, particularly for migratory species [42]. We could quantify such costs through behavioral methods. Hence, we could gain a more comprehensive understanding of the influence of in-flight collisions hazards on particular species (e.g. energetic costs vary by body size) and expand our understanding of how landscapes and temporal factors influence the costs of collision hazards.

Although we did not achieve species-level identifications in this study, we did note the general size of birds and this factor influenced some model outcomes. In general, smaller birds showed greater changes in flight metrics (S3, S5 and S6 Figs). We explain this by flight kinematic differences according to body size. Smaller birds can produce greater mass-specific muscle power than larger birds, and can therefore more readily adjust their flight behaviors [59, 60]. Interestingly, there appears to be some interaction between stimulus type (4–6 vs 6–8 kHz) and bird size such that differences in behaviors by bird size appear to be more pronounced during 6–8 kHz treatment conditions compared to 4–6 kHz treatment conditions. As the 6–8 kHz signal likely led to delayed responses, birds in this treatment had less space and time to respond and small birds were more able to adjust their flight under these constraints compared to larger birds. This pattern could also be affected by taxonomic differences in auditory sensitivities. In general, larger avian taxa have poorer hearing at higher frequencies [47, 49, 50].

The date of data collection and the tower site location were notable factors influencing flight behaviors. However, sampling date did not significantly influence the effect of treatment condition on flight behavior in any of our models (Tables 2–7). Differences in flight behavior between sampling dates may be attributable to a range of factors including weather conditions (though we did not detect strong associations between our measured weather variables and

flight behavior) or differences in the assemblage of birds moving through the area on different dates, as is common during migrations. Tower site location did influence the effect of treatment condition on flight behaviors (Tables 3 and 5). There are many local factors which could influence the flight behaviors of birds around towers, such as surrounding landcover type and the taxonomic makeup of local birds, particularly resident individuals. Of particular interest would be whether resident individuals differed in their behavioral responses to sound stimuli from non-resident individuals. We were unable to test this in the current study, but future work should explore this potential source of variation further, as it could begin to address whether birds may habituate to acoustic warning signals associated with collision hazards.

The implementation of acoustic deterrence methods in open-air contexts may be relatively accessible. For example, sound sources may be mounted on or near structures, using highly directional sound fields to minimize potential noise pollution, as was done in our study. Future work should also investigate any differential influence of the placement of the sound source relative to the hazardous structure. Our study, due to logistical constraints, placed directional speakers at the base of towers. However, previous work has illustrated the prevalence of ecologically referential signals in nature. For example, studies have shown that signal receivers direct attention in a spatially appropriate manner in response to certain types of alarm calls, such as directing visual attention upwards in response to alarm signals specific to aerial predators [61, 62]. Other research has shown, more generally, the tendency of multiple species to orient visual attention based partly on simple signal characteristics such as frequency [63]. Collision mitigation approaches could co-opt such natural tendencies in the behaviors of at-risk birds, to help elicit collision avoidance. Conversely, there could be unintended consequences of using signals familiar to wildlife, such as attraction to rather than deterrence from hazards. In such instances, novel and unfamiliar signals may prove more effective.

In general, the use of acoustic signals in mitigating collisions with open-air hazards may be more appropriate in some settings than others. Given the similarities between avian and human auditory sensitivities [47], the use of acoustic signals in areas close to humans may result in unwanted nuisance noise. Acoustic warning signals could also present challenges to other wildlife, through masking of communication signals or increasing stress through a variety of mechanisms [64]. Some geographical areas may be more suitable than others based on their community composition and any implementation of acoustic warning methods should pay careful attention to the makeup of and potential impacts on local wildlife populations. To reduce unintended negative consequences of acoustic warning signals, context-dependent intermittent use may be a compelling solution. For example, signals may only be projected during higher risk periods such as at times of peak migration, under certain weather conditions that have been associated with elevated collision risk, or when birds are detected in the area through motion detection or radar technology. As with any mitigation approach, the use of acoustic warning signals should be tailored to the context of a given hazard, including its location and surrounding ecological communities, the predominant environmental conditions of the area, and the characteristics of any at-risk bird populations.

Overall, it is clear that the 4–6 kHz acoustic signal we deployed decreased collision risk. The use of acoustic signals in mitigating collisions in open airspace thus merits further exploration. We tested the acoustic lighthouse concept in one context—flights around tall communication towers during fall migration. There would be great value in extending this test to other times of year, to other landscapes and seascapes, and to other structures that present collision hazards. In particular, we feel the acoustic lighthouse has potential to reduce in-flight collision risks associated with wind turbines, both on- and offshore, especially since visual attention has been shown to influence birds' probability of collisions with wind turbine blades [20]. The acoustic lighthouse could augment previous approaches to increase detectability and further

reduce collisions and avian mortality, leading to reduced conflict between economic development and wildlife conservation.

## Supporting information

**S1 Fig. Example flight and derived behavioral metrics.** The top row of graphs illustrates a smoothed 3D reconstruction of a bird flight path around a tower. Flight behavior was characterized using measures of horizontal distance from the tower (d), absolute velocity (v), and horizontal displacement angle from the tower ($\theta_{tower}$). These measures were summarized for an entire flight path using the median. Changes in flight behavior over the course of a bird's flight were summarized using the change in the median from the earlier to latter half of the bird's flight.
(TIF)

**S2 Fig. Change in distance by treatment condition and tower site location.** Change in distance by treatment condition within each tower site. Solid red dots and error bars represent model-averaged mean of outcome variable ± standard error of the mean. Green dotted line indicates a level of change in distance where flights remain further away from the tower. Orange dotted line indicates a level of change in distance where flights draw closer to the tower.
(TIF)

**S3 Fig. Change in distance by treatment condition and bird size.** Solid red dots and error bars represent model-averaged mean of outcome variable ± standard error of the mean. Symbols indicate the attribute of bird size for each data point.
(TIF)

**S4 Fig. Change in velocity by treatment condition and tower site location.** Change in velocity by treatment condition within each tower site. Solid red dots and error bars represent model-averaged mean of outcome variable ± standard error of the mean. Green dotted line indicates the level of change in velocity representing a maintaining of flight speed. Orange dotted line indicates the level of change in velocity representing an acceleration in flight speed.
(TIF)

**S5 Fig. Change in velocity by treatment condition and bird size.** Solid red dots and error bars represent model-averaged mean of outcome variable ± standard error of the mean. Symbols indicate the attribute of bird size for each data point.
(TIF)

**S6 Fig. Change in angle of displacement by treatment condition and bird size.** Solid red dots and error bars represent model-averaged mean of outcome variable ± standard error of the mean. Symbols indicate the attribute of bird size for each data point.
(TIF)

**S1 Table. Set of candidate models.** The same set of candidate models was applied to all outcome flight behavior metrics. Structure of linear predictors was based on a-priori hypotheses and exploration of non-linearity between predictors and response variables and of collinearity between predictor variables.
(DOCX)

**S2 Table. Overall distance final model set.** AICc weight was used to rank model suitability. Models carrying 95% of total AICc weights were preserved and worse performing but more

complex nested models were removed.
(DOCX)

**S3 Table. Change in distance final model set.** AICc weight was used to rank model suitability. Models carrying 95% of total AICc weights were preserved and worse performing but more complex nested models were removed.
(DOCX)

**S4 Table. Overall velocity final model set.** AICc weight was used to rank model suitability. Models carrying 95% of total AICc weights were preserved and worse performing but more complex nested models were removed.
(DOCX)

**S5 Table. Change in velocity final model set.** AICc weight was used to rank model suitability. Models carrying 95% of total AICc weights were preserved and worse performing but more complex nested models were removed.
(DOCX)

**S6 Table. Overall angle of displacement final model set.** AICc weight was used to rank model suitability. Models carrying 95% of total AICc weights were preserved and worse performing but more complex nested models were removed.
(DOCX)

**S7 Table. Change in angle of displacement final model set.** AICc weight was used to rank model suitability. Models carrying 95% of total AICc weights were preserved and worse performing but more complex nested models were removed.
(DOCX)

**S8 Table. Weather variables across sampling dates.** Precipitation (mm), mean temperature (˚C) and cloud cover (oktas) are reported. Average daily estimates of weather variables were extracted from the PRISM Climate Group gridded dataset (Oregon State University).
(DOCX)

**S1 Appendix. Bird activity data.** Raw data on bird activity and flight classification around towers.
(XLSX)

**S2 Appendix. Flight metrics data.** Measures of flight behavior derived from reconstructed positional data of all "at-risk" bird flights around towers.
(XLSX)

**S1 Audio. 4–6 kHz acoustic stimulus.** The 4–6 kHz acoustic stimulus was generated by band-pass filtering white noise between 4 and 6 kHz. Audio files were generated using PRAAT (version 6.0.46) and edited in Audacity (R) (version 2.4.1).
(MP3)

**S2 Audio. 6–8 kHz acoustic stimulus.** The 6–8 kHz acoustic stimulus was generated by band-pass filtering white noise between 6 and 8 kHz. Audio files were generated using PRAAT (version 6.0.46) and edited in Audacity (R) (version 2.4.1).
(MP3)

## Acknowledgments

We thank the Delmarva Educational Association and the Virginia Department of State Police for access to communication tower sites, as well as Glenn Hickman from WHRO Public Media for study site recommendations. We thank Dan Cristol and Matthias Leu for comments on early versions of this manuscript. We thank Sam Mason for feedback on statistical analyses and Kennedy O'Neil, Alex Mooney, and Caitlyn Marat for their work on data digitization.

## Author Contributions

**Conceptualization:** Timothy J. Boycott, John P. Swaddle.

**Data curation:** Timothy J. Boycott.

**Formal analysis:** Timothy J. Boycott, Sally M. Mullis.

**Funding acquisition:** Timothy J. Boycott, John P. Swaddle.

**Investigation:** Timothy J. Boycott, Sally M. Mullis.

**Methodology:** Timothy J. Boycott, Sally M. Mullis, Brandon E. Jackson, John P. Swaddle.

**Project administration:** John P. Swaddle.

**Resources:** Brandon E. Jackson, John P. Swaddle.

**Software:** Brandon E. Jackson.

**Supervision:** John P. Swaddle.

**Validation:** Brandon E. Jackson, John P. Swaddle.

**Writing – original draft:** Timothy J. Boycott, John P. Swaddle.

**Writing – review & editing:** Timothy J. Boycott, Sally M. Mullis, Brandon E. Jackson, John P. Swaddle.

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
