## [Decision Letter · Decision Letter 0]

9 Feb 2021

PONE-D-21-00147

Field testing an “acoustic lighthouse”: Combined acoustic and visual cues provide a multimodal solution that reduces avian collision risk with tall human-made structures

PLOS ONE

Dear Dr. Boycott,

Thank you for submitting your manuscript to PLOS ONE. After careful consideration, we feel that it has merit but does not fully meet PLOS ONE’s publication criteria as it currently stands. Therefore, we invite you to submit a revised version of the manuscript that addresses the points raised during the review process.

Please submit your revised manuscript in the next 20 days. If you will need more time than this to complete your revisions, please reply to this message or contact the journal office at plosone@plos.org. Please include the following items when submitting your revised manuscript:

We look forward to receiving your revised manuscript.

Kind regards,

Deborah Faria, PhD

Academic Editor

PLOS ONE

Additional Editor Comments:

Dear Dr. Boycott,,

I have received the reports from the advisors on your manuscript entitled " Field testing an “acoustic lighthouse”: Combined acoustic and visual cues provide a multimodal solution that reduces avian collision risk with tall human-made structures", which you submitted to the PlosOne.

Based on the reviews received, your manuscript could be considered for publication pending the incorporation of minor revisions. For this reason, you are asked to carefully consider the comments of both reviewers.

I am looking forward to receiving your revised manuscript within 20 days. Should you need more time to accomplish the revision please do not hesitate to inform me immediately.

Journal Requirements:

3. We note that you have a patent relating to material pertinent to this article. Please provide an amended statement of Competing Interests to declare this patent (with details including name and number), along with any other relevant declarations relating to employment, consultancy, patents, products in development or modified products etc. Please confirm that this does not alter your adherence to all PLOS ONE policies on sharing data and materials, as detailed online in our guide for authors http://journals.plos.org/plosone/s/competing-interests by including the following statement: "This does not alter our adherence to  PLOS ONE policies on sharing data and materials.” If there are restrictions on sharing of data and/or materials, please state these. Please note that we cannot proceed with consideration of your article until this information has been declared.

Reviewers' comments:

Reviewer's Responses to Questions

**Comments to the Author**

1. Is the manuscript technically sound, and do the data support the conclusions?

Reviewer #1: Partly

Reviewer #2: Yes

2. Has the statistical analysis been performed appropriately and rigorously? 

Reviewer #1: Yes

Reviewer #2: Yes

3. Have the authors made all data underlying the findings in their manuscript fully available?

Reviewer #1: Yes

Reviewer #2: Yes

4. Is the manuscript presented in an intelligible fashion and written in standard English?

Reviewer #1: Yes

Reviewer #2: Yes

5. Review Comments to the Author

Reviewer #1: Overall, the methods of this study were rigorous. However, the sample size is quite small, both in terms of the number of days during which data were collected and in the number of "at risk" flights.

The sample size of "at risk" flights was 145. State that up front in the abstract. I'm not convinced that the data supports the claim, "We recorded an overall 12-16% reduction in bird activity surrounding towers during sound treatment conditions, compared with control (no broadcast sound) conditions." As stated on p. 19, lines 377-8, the rate of "at risk" interactions did not vary substantially between treatment protocols. Flight behavior during different treatments should only be compared using observations from the same day. As noted, the sampling date was a dominant factor. I suggest modifying the conclusions accordingly.

Reviewer #2: This an interesting ms dealing with test multimodal systems to avoid or decrease bird collisions with towers. The original approach used is an interesting contribution for other similar experiments in the same or different infrastructures, saving time and avoiding well-known limitations in field works based on victims counts. I have only some comments:

• L 75: I suggest to include some references to mitigation devices in power lines “…and marking ground wires in transmission power lines (Ferrer, M., Morandini, V., Baumbush, R., Muriel, R., De Lucas, M., & Calabuig, C. 2020. Efficacy of different types of “bird flight diverter” in reducing bird mortality due to collision with transmission power lines. Global Ecology and Conservation, e01130).

• Would be useful and nice if you can include a link to hear the sound you used

• Table 1: Unnecessary and difficult to understand as it is now.

• I would like to see in Discussion section something about habituation as a limiting factor to use this approach. Working with migratory birds during migration probably does no room for habituation but in sedentary birds would be different.

6. PLOS authors have the option to publish the peer review history of their article (what does this mean?). If published, this will include your full peer review and any attached files.

Reviewer #1: No

Reviewer #2: **Yes: **Miguel Ferrer

---

## [Author Response · Author response to Decision Letter 0]

28 Feb 2021

Dear Dr. Faria,

We would like to thank you and both of the reviewers for your feedback and constructive comments on our initial submission of this manuscript. We have responded to all comments and have made appropriate alterations to the manuscript and accompanying information. We believe these revisions have strengthened the paper, for which we are grateful. 

Response to Academic Editor, Deborah Faria, PhD

1. We have ensured that the manuscript and all associated files meet PLOS ONE’s style requirements, including manuscript formatting and file naming. We have re-formatted some in-text names of supplemental tables (lines 420, 459) and we have edited the file names for all figures and tables (including supporting information files) to exclude titles and have re-uploaded these files to the Editorial Manager. 

2. We have updated our ‘Funding Information’ to include the same grant information provided in the ‘Financial Disclosure’. We have included grant numbers where applicable. Several of the funding sources do not have external grant numbers associated with these awards. This information has also been provided in our updated cover letter. 

Financial Disclosure/Funding Information: This work was supported by the Center for Innovative Technology’s Commonwealth Research Commercialization Fund award MF18-029-En (https://www.cit.org/) to JPS, by the Animal Welfare Institute (https://awionline.org/), the Virginia Society of Ornithology (https://www.virginiabirds.org/), the Williamsburg Bird Club (http://williamsburgbirdclub.org/), and the Department of Arts and Sciences at William & Mary to TJB. The funders had no role in study design, data collection and analysis, decision to publish, or preparation of the manuscript.

3. We have amended our statement of competing interests to more comprehensively declare a provisional patent relating to this work on which two authors are listed. This statement confirms that affiliation with this patent does not alter our adherence to all PLOS ONE policies on sharing data and materials. This information has also been provided in our updated cover letter.

Competing Interests: The authors have read the journal's policy and the authors of this manuscript have the following competing interests: The work reported here contributed to US Provisional Application No. 63/082,025 (Swaddle, J. P. and Boycott, T, J, 2020. Systems and methods for reducing the risks of bird strike), on which Timothy Boycott and John Swaddle are listed as inventors. This patent could be commercialized in the future. This does not alter our adherence to PLOS ONE policies on sharing data and materials.

4. We have updated our Data Availability statement following feedback from both reviewers that all data underlying our findings have been made fully available. All data underlying the analyses and results presented in the manuscript are included as supplemental information files. This information has also been provided in our updated cover letter.

Data Availability: Yes - all data are fully available without restriction.

All relevant data are within the manuscript and its Supporting Information files.

Response to Reviewer #1

Reviewer: Overall, the methods of this study were rigorous. However, the sample size is quite small, both in terms of the number of days during which data were collected and in the number of "at risk" flights.

Response: We thank the reviewer for their support of the methods used in our study. We do recognize that the number of days over which our data were collected are relatively few. However, we believe that a particular strength of our study is the robust sample size of bird interactions generated from this relatively short sampling time. Many other studies of avian mortality at human-made structures produce similar, or even smaller, sample sizes of bird measures from much longer sampling efforts (e.g. Gehring et al. 2009; Barrios and Rodríguez 2004; Klem and Saenger 2013; reviewed in Pagel et al. 2013; Loss et al. 2014). However, we do recognize the strong effect that sampling date will consequently have on our results, given the relatively smaller sampling effort. Therefore, as described below, we have strengthened our discussion on the influence of sampling date on our results (lines 689-705). 

Reviewer: The sample size of "at risk" flights was 145. State that up front in the abstract.

Response: This information has been added to the abstract, as suggested by the reviewer (line 42)

Reviewer: I'm not convinced that the data supports the claim, "We recorded an overall 12-16% reduction in bird activity surrounding towers during sound treatment conditions, compared with control (no broadcast sound) conditions." As stated on p. 19, lines 377-8, the rate of "at risk" interactions did not vary substantially between treatment protocols.

Response: While the proportion of all detected birds that were categorized as having “at-risk” flights did not differ substantially between treatments, the overall number of birds within the general vicinity of towers did differ between treatment conditions. We believe that this is important information to include. The measure of bird activity around towers was generated from scoring bird activity within roughly 500 m of towers. It is possible that treatment conditions influenced bird activity beyond this distance such that the number of birds entering within 500 m of towers was influenced by the presence or absence of the acoustic stimuli. However, we recognize that our data do not allow us to demonstrate this potential effect. Therefore, we have altered the language used when discussing these data to place greater emphasis on the potential effect that treatment conditions could have on the general bird activity rates around the broader vicinity of towers, and the more compelling effect that treatments do have on bird flight behavior nearer to the towers. For example, when comparing activity rates between treatments, we have replaced words such as “reduction” with words such as “lower” to reflect the correlational rather than causational nature of these particular data (Abstract: lines 39-40; Results: lines 360, 362, 364-366; Discussion: lines 578-584). 

Reviewer: Flight behavior during different treatments should only be compared using observations from the same day. As noted, the sampling date was a dominant factor. I suggest modifying the conclusions accordingly.

Response: We recognize that the reviewer does not suggest revisions to our data or statistical analyses, but rather some revisions to the discussion and conclusions drawn from these analyses. We believe that direct comparisons of flight metrics between treatment conditions, across all our sampling conditions, is central to the purposes of this study. Our statistical approach allows for this comparison, by accounting for variation in flight metrics due to variation in sampling date. We acknowledge the reviewer’s concern for the smaller sample size of sampling date, but we feel confident in having accounted for this aspect of our study design in our statistical approach. However, as stated above, we recognize that the importance of sampling date for the variation in flight behavior warrants more discussion of this effect in our manuscript. We have added a section to our discussion and have also included a brief discussion of the effect of tower location on flight behavior, since this variable was another dominant factor influencing results. At this point, we also discuss the potential role of habituation behavior, as suggested by Reviewer 2 (lines 689-705). 

Response to Reviewer #2, Miguel Ferrer

Reviewer: This an interesting ms dealing with test multimodal systems to avoid or decrease bird collisions with towers. The original approach used is an interesting contribution for other similar experiments in the same or different infrastructures, saving time and avoiding well-known limitations in field works based on victims counts.

Response: We thank the reviewer for their encouraging description of our study and methods, particularly given their own experience assessing avian collision mortality and mitigation measures. 

Reviewer: L 75: I suggest to include some references to mitigation devices in power lines “…and marking ground wires in transmission power lines (Ferrer, M., Morandini, V., Baumbush, R., Muriel, R., De Lucas, M., & Calabuig, C. 2020. Efficacy of different types of “bird flight diverter” in reducing bird mortality due to collision with transmission power lines. Global Ecology and Conservation, e01130).

Response: We thank the reviewer for their suggestion of this recent reference. We agree that the study described is a compelling example of an effective collision mitigation technique and therefore important to include in the introductory section of our manuscript. The reference has been included along with citations of other studies which employ visual signals to reduce the incidence of avian collisions with human-made structures (line 76). 

Reviewer: Would be useful and nice if you can include a link to hear the sound you used

Response: We have included short (5 second) examples of both acoustic stimuli types (4-6 and 6-8 kHz band-pass filtered white noise) as supporting information files (line 173). These files would be hosted on PLOS’ servers and accessible via hyperlinks in the online version of the published article. These files have been uploaded to the Editorial Manager and captions have been added to the Supporting information section of the manuscript (lines 1031-1039).

Reviewer: Table 1: Unnecessary and difficult to understand as it is now.

Response: We agree with the reviewer that this table is excess information and is difficult to understand in its current format. We have removed Table 1 and believe that the in-text description of our general study design is sufficient in effectively conveying this information (lines 179-181; lines 210-217). All remaining tables, legends and their in-text citations have consequently been renumbered. 

Reviewer: I would like to see in Discussion section something about habituation as a limiting factor to use this approach. Working with migratory birds during migration probably does no room for habituation but in sedentary birds would be different.

Response: We agree with the reviewer that this is an important behavioral element to include in the discussion of acoustic warning signals as collision deterrents. We have added a section to our discussion which focuses on some of the dominant factors which influenced our results, including sampling date, location, and bird assemblage. We found this to be an appropriate place to bring up the potential role which habituation may play in the efficacy of this approach (lines 689-705). 

Additional revisions:

We have also updated the acknowledgements section to correct a previous omission (lines 761-763).

We hope that these alterations address all the comments raised by yourself and the reviewers and that you find our revised manuscript suitable for publication in PLOS ONE. We look forward to your final decision. 

Sincerely,

Timothy Boycott

Biology Department

William & Mary

---

## [Editor Report · Decision Letter 1]

26 Mar 2021

Field testing an “acoustic lighthouse”: Combined acoustic and visual cues provide a multimodal solution that reduces avian collision risk with tall human-made structures

PONE-D-21-00147R1

Dear Dr. Boycott,

We’re pleased to inform you that your manuscript has been judged scientifically suitable for publication and will be formally accepted for publication once it meets all outstanding technical requirements.

Kind regards,

Deborah Faria, PhD

Academic Editor

PLOS ONE

Additional Editor Comments (optional):

Dear Dr. Boycott,

Regarding the manuscript “Field testing an “acoustic lighthouse”: Combined acoustic and visualcues provide a multimodal solution that reduces avian collision risk with tallhuman-made structures” I received your review and it is my understanding that all concerns raised by the reviewers were adequately addressed, and therefore I consider that the manuscript can be accepted forpublication. All the accepted suggestions improved considerably the manuscript. It is also worth mentioning that, in this version, you fully attended to specific requirements by the PlosOne, including manuscript formatting and file naming, statements of funding information and financial disclosure, more detailed information regarding the statement of competing interests, and declaration of data availability.

Thanks again for publishing inPlosOne.

Yours sincerely

Dr. Deborah Faria
---

## [Editor Report · Acceptance letter]

5 Apr 2021

PONE-D-21-00147R1 

Field testing an “acoustic lighthouse”: Combined acoustic and visual cues provide a multimodal solution that reduces avian collision risk with tall human-made structures 

Dear Dr. Boycott:

I'm pleased to inform you that your manuscript has been deemed suitable for publication in PLOS ONE. Congratulations! Your manuscript is now with our production department. 

Kind regards, 

on behalf of

Dr. Deborah Faria 

Academic Editor

PLOS ONE